# Development of an Australian Bread Wheat Nested Association Mapping Population, A New Genetic Diversity Resource for Breeding under Dry and Hot Climates

**DOI:** 10.3390/ijms22094348

**Published:** 2021-04-21

**Authors:** Charity Chidzanga, Delphine Fleury, Ute Baumann, Dan Mullan, Sayuri Watanabe, Priyanka Kalambettu, Robert Pontre, James Edwards, Kerrie Forrest, Debbie Wong, Peter Langridge, Ken Chalmers, Melissa Garcia

**Affiliations:** 1School of Agriculture, Food and Wine, The University of Adelaide, Glen Osmond, SA 5064, Australia; charity.chidzanga@adelaide.edu.au (C.C.); delphine.fleury@adelaide.edu.au (D.F.); ute.baumann@adelaide.edu.au (U.B.); sayuri.watanabe@adelaide.edu.au (S.W.); priyanka.kalambettu@adelaide.edu.au (P.K.); peter.langridge@adelaide.edu.au (P.L.); kenneth.chalmers@adelaide.edu.au (K.C.); 2ARC Industrial Transformation Research Hub for Wheat in a Hot and Dry Climate, Waite Research Institute, The University of Adelaide, Glen Osmond, SA 5064, Australia; DMullan@intergrain.com (D.M.); James.Edwards@agtbreeding.com.au (J.E.); 3Intergrain 19 Ambitious Link, Bibra Lake, WA 6163, Australia; RPontre@intergrain.com; 4Australian Grain Technologies, 20 Leitch Rd, Roseworthy, SA 5371, Australia; 5Genomics & Cell Sciences, Agriculture Victoria Research, Department of Jobs, Precincts and Regions, Agribio, 5 Ring Rd, Bundoora, VIC 3083, Australia; kerrie.forrest@agriculture.vic.gov.au (K.F.); debbie.wong@agriculture.vic.gov.au (D.W.)

**Keywords:** nested association mapping, genetic diversity, exotic germplasm, wheat, multi-environmental trials, QTL

## Abstract

Genetic diversity, knowledge of the genetic architecture of the traits of interest and efficient means of transferring the desired genetic diversity into the relevant genetic background are prerequisites for plant breeding. Exotic germplasm is a rich source of genetic diversity; however, they harbor undesirable traits that limit their suitability for modern agriculture. Nested association mapping (NAM) populations are valuable genetic resources that enable incorporation of genetic diversity, dissection of complex traits and providing germplasm to breeding programs. We developed the OzNAM by crossing and backcrossing 73 diverse exotic parents to two Australian elite varieties Gladius and Scout. The NAM parents were genotyped using the iSelect wheat 90K Infinium SNP array, and the progeny were genotyped using a custom targeted genotyping-by-sequencing assay based on molecular inversion probes designed to target 12,179 SNPs chosen from the iSelect wheat 90K Infinium SNP array of the parents. In total, 3535 BC_1_F_4:6_ RILs from 125 families with 21 to 76 lines per family were genotyped and we found 4964 polymorphic and multi-allelic haplotype markers that spanned the whole genome. A subset of 530 lines from 28 families were evaluated in multi-environment trials over three years. To demonstrate the utility of the population in QTL mapping, we chose to map QTL for maturity and plant height using the RTM-GWAS approach and we identified novel and known QTL for maturity and plant height.

## 1. Introduction

The expression of most traits of agronomic importance in crop species vary in degree and are determined by the cumulative effect of multiple interacting genetic loci and the environment [1]. The underlying genetic loci for quantitative traits can be dissected through QTL mapping to provide valuable insights into the genetic architecture of these traits [2]. Understanding the genetic architecture of quantitative traits such as yield is important in the development of improved crop varieties [3]. In the past, QTL mapping studies have been performed through linkage analysis in segregating populations developed from bi-parental crosses [4]. In the development of bi-parental mapping populations, the number of recombination events is limited, and the amount of genetic diversity is restricted to that which is present in the two parents [5]. As a result, QTL mapping in bi-parental populations, although successful in detecting numerous QTL in various crops [6,7,8,9], is constrained by low resolution, lack of genetic diversity and potentially an inability to detect the same QTL in other germplasm.

To compensate for the limitations of QTL mapping in bi-parental populations, genome-wide association mapping (GWAS) has been utilised [10]. GWAS populations consist of diversity panels that may comprise collections of wild lines, landraces, released varieties and breeding lines. GWAS makes use of historical recombination events and hence captures more allelic diversity and maps QTL with greater resolution [11,12]. GWAS is, however, limited by the confounding effects of inherent population stratification and cryptic relatedness that results in spurious associations [13]. Different statistical methods have been developed to control the effect of population stratification and relatedness [14,15] but these methods also result in an increase in false negative associations as they have a reduced power to detect QTL associated with population structure [16].

In order to combine the strengths of bi-parental and association mapping, a Nested Association Mapping (NAM) approach was developed [17]. A NAM population offers increased allelic diversity, samples of both recent and historic recombination events, and reduces the confounding effect of population structure [18]. A NAM population comprises a large set of related progenies within multiple bi-parental populations developed by selecting a diverse set of founder lines and crossing them to a common reference parent. Founder lines in a NAM are ideally chosen to maximize genetic diversity and can, therefore, include exotic germplasm, wild relatives and landraces. The reference parent is usually a well-characterised, locally adapted elite line [17]. The incorporation of genetically diverse founder lines introduces novel genetic diversity, while crossing the founder lines with a common reference line balances the need for genetic diversity with elite performance. The NAM design results in progeny genomes that are a mix of chromosome segments from either the diverse founder lines or the common parent [17]. The reshuffling of parental genomes breaks down population structure while introducing recent recombination events [17]. First developed in maize genetics studies, the NAM design has been a successful tool in dissecting the genetic architecture of traits in various crops [19,20,21,22].

Drought and heat are permanent features of the Australian climate, making it the driest inhabited continent on Earth [23,24]. Wheat production in Australia is mostly rainfed and, therefore, drought and heat are regular and serious abiotic stresses limiting wheat productivity. Wheat yields in Australia have stalled since 1990 due to drought and rising temperatures. The severity and frequency of these stresses are expected to increase with changes in global climate. Breeding wheat for varieties with stable yields under these stress conditions is an imperative. However, due to the complex nature of drought and heat stress tolerance in wheat, it is difficult to achieve this breeding goal. Furthermore, success in breeding requires the availability of genetic diversity, knowledge of the genetic architecture of the traits of interest [25] and efficient means of transferring the desired genetic diversity into the relevant genetic background. In wheat genetics, the effort to understand the genetics of heat and drought tolerance have occurred separately through QTL mapping in bi-parental and association mapping populations [26,27,28]. Although these efforts have yielded useful information on the genetic architecture of these traits, they have not been able to simultaneously identify diverse favourable alleles for both and develop lines that can be readily introduced into breeding programs. NAM populations can simultaneously map QTL and provide useful germplasm for breeding programs. The shuffling of genomes during the development of NAM lines results in a subset of lines with new allele combinations that can cause them to outperform their parents [29]. Since the common parent in NAM populations is usually well characterized and well adapted, any NAM line that outperforms the common parent can be considered potential breeding material [30].

Many reported NAM populations are designed to have only one common parent crossed to a number of diverse founder lines. Here, we report the development of an ARC Industrial Transformation Research Hub for Wheat in a Hot and Dry Climate NAM population (here onwards referred to as the OzNAM) using two common reference parents. While the main goal for developing the OzNAM is to enhance wheat breeding under dry and hot climates, here we present data on the genetic diversity, phenotypic evaluation and mapping of QTL associated with maturity and plant height in a subset of the OzNAM population so as to demonstrate the utility of the population in QTL mapping. Both maturity and plant height are controlled by major well characterized loci as well as other regions of smaller effect. Furthermore, these two traits are correlated with yield and are strongly selected in plant breeding.

## 2. Results

### 2.1. Development of the OzNAM Population

For the development of the OzNAM population, we initially started with a selection of 76 exotic parents; however, due to unsuccessful F_1_ crosses, naturally occurring bushfires, and delays in seed processing, a number of potential NAM families were excluded from the population. In total, 125 crosses between the exotic parents and the two common reference parents were successfully advanced to BC1F_4:6_ (Appendix A). The resulting NAM population includes 3535 BC1F_4:6_ RILs from 125 families. Sixty-seven families have Gladius as the common reference parent and 58 families have Scout as the common reference parent. The number of lines per NAM family ranges from 21 to 76 lines.

### 2.2. Phenotypic Data Analysis

In this paper, we present phenotypic BLUPs and GWAS results from a subset of the OzNAM population field trialled at four sites over a three-year period across the Australian wheat belt. Figure 1 and Figure 2 show variation in Zadoks’ score [31] and plant height in different sites, respectively. In all the field trials, maturity and plant height showed broad phenotypic variability between and within families (Appendix A). The mean value of Zadoks’ score ranged from 54 to 62 per trial. Average plant height per trial ranged from 77 to 97 cm. The heritability for the two traits was very high with H^2^ estimates ranging from 0.8 to 0.95 (Table 1). At least 77% of the lines in the 2018 and 2019 trials had Zadoks’ scores that were within the selection thresholds and 94% of the lines had plant height measurements below 115 cm. This indicates the effectiveness of the selection based on the 2017 data. The Dandaragan 2018 trial, being located at a different location from the site at which selection thresholds were based on, had 23% of the lines outside the selection thresholds for Zadoks’ score.

### 2.3. Genotypic Data and Population Structures

A total of 4964 polymorphic and multi-allelic markers that spanned the whole genome were used in this study: 2189 markers mapped to the A genome; 2142 markers mapped to the B genome; only 581 markers mapped to the D genome; 52 markers mapped to the RefSeq v1.0 unallocated part of the genome. The called alleles per marker ranged from two to 29 with an average of 5.6 alleles per marker. The called alleles ranged from 2–26 in the A genome, 2–29 in the B genome, and 2–19 in the D genome.

### 2.4. Introgression and Population Structure

3494 and 3435 genome-wide homozygous polymorphic markers were identified for Gladius- and Scout-derived subpopulations, respectively. The overall genome wide proportions of exotic alleles across all families within the Gladius and Scout subpopulations were 27.7% and 21.9%, respectively. Within individual lines, the proportion of exotic alleles ranged from 7.3–61.5% in the Gladius derived subpopulation and from 5.5–59.8% in the Scout derived lines. Up to 42% of individual chromosomes is represented by alleles from exotic parents. Variation in the amount and distribution of exotic alleles between chromosomes and subpopulations is apparent (Appendix A). Figure 3 shows variations in parental allelic contributions between subpopulations on selected chromosomes. Chromosome 2D carries the photoperiod-response *Ppd-D1* gene [32], while chromosomes 4B and 4D carry the reduced height *Rht-B1* and *Rht-D1* genes [33]. These genes are crucial for the adaptability of wheat in various environments and certain alleles could have been selected through our culling for maturity, plant height and yield during the development of the population. Chromosome 1 represents parental allelic contributions across the genome where there is no known selection pressure. The Gladius-derived subpopulation shows a higher proportion (0.68) of the exotic alleles around the photoperiod *Ppd-D1* gene compared to the Scout-derived subpopulation (0.03). Variations in introgression between the two subpopulations is apparent along the chromosome, with higher proportions of the exotic alleles at the end of the chromosome in both populations. In the region adjacent to the *Rht-D1* loci, Scout-derived families had an exotic allele frequency of 0.4 which is higher than the expected 0.25. For Gladius-derived families, there were no markers close enough to the *Rht-D1* locus. This gap in markers was due to the filtering step where we removed heterozygous markers. For the *Rht-B1* proximal region, the average exotic allele frequency for the Scout derived families was 0.26 and 0.14 for the Gladius derived families.

Population structure evaluation for the whole NAM population combined with the parents revealed three distinct groups: the exotic parents, the Gladius families and the Scout families. As expected, the reference parents clustered with their respective families. The first eigenvector explained 64.3% of the variation while the second and third eigenvectors explained 9.0% and 1.5%, respectively (Figure 4). There was a slight overlap between a few of the Gladius and Scout lines, with each other and with the exotic parents. This could be explained by the unintentional mislabelling of some Gladius lines as Scout lines and vice versa. The overlap with the exotic parents may be attributed to accidental selfing during population development.

### 2.5. QTL Mapping in the NAM Population

We mapped QTL associated with maturity and plant height in a subset of the NAM population using phenotypic BLUPs from each of the field trial environments separately. For both traits, significant marker-trait associations were detected in all the trial environments with the highest number of QTL detected in the Roseworthy 2017 trial. For maturity, a single highly significant locus associated with the photoperiod response *Ppd-D1* locus on chromosome 2D was detected in all four environments (Table 2). An additional four loci associated with maturity were also detected: two loci on chromosomes 2B and 3A were detected in the Roseworthy 2017 trial while another two, on chromosome 5A and 3A, were detected in the Roseworthy 2018 and Dandaragan 2018 trial, respectively. The five loci detected had 2 to 5 different alleles. The amount of variation explained by all detected loci ranged from 3.2% to 34.3% with the *Ppd-D1* locus explaining the highest amount of variation in each of the environments.

For plant height, a single highly significant locus associated with the *Rht-D1* locus on chromosome 4D was detected in all trial environments (Table 2). An additional highly significant locus on chromosome 4B associated with the *Rht-B1* locus was also detected in the Roseworthy 2017 trial. Another locus with 11 alleles was detected on chromosome 6B in the Dandaragan 2018 trial. The *Rht-D1* associated locus explained the highest amount of variation in all environments compared to the other detected loci.

## 3. Discussion

Exotic germplasm are reservoirs of useful genetic diversity for traits of agronomic importance such as drought and heat stress tolerance [34]. Utilization of exotic germplasm in genetic studies and plant breeding offers opportunities for understanding the genetic architecture of traits and the improvement of agronomic performance of modern crop varieties [35]. However, exotic germplasm carries a range of undesired traits, which limit their suitability for modern agriculture. In this study, we developed a NAM population that incorporated genetic diversity from exotic germplasm. The rationale for developing our NAM population was to improve the chances of success in incorporating alleles from exotic germplasm into Australian breeding programs to improve drought and heat tolerance. There was a need to develop a resource for wheat genetics and breeding in Australia that could (i) increase the chances of success in incorporating novel alleles from exotic germplasm into Australian material; (ii) enable easy utilization of exotic germplasm in high resolution QTL mapping; (iii) enable the transfer of knowledge and breeding material to Australian breeding programs.

Since the development and implementation of the first NAM population in maize, the NAM design has been adopted in many other crops. It has been useful in giving insight into the genetic architecture of complex traits, such as disease resistance, flowering time and plant height amongst other traits [18,22,36,37,38]. Most NAM populations have been designed to have just one reference parent and several donor parents. Our NAM population has two elite Australian wheat varieties as the reference parents. Choosing these two as reference parents ensured the generation of a genetic mapping population that can be widely grown across different growing regions and evaluated for many agronomic traits. Having two reference parents also allows for the creation of parallel families that can mirror each other, and therefore be compared to validate QTL. It also increases the population size and the chances of successfully introgressing favourable exotic alleles into Australian germplasm.

Phenotypic evaluations revealed diversity in the NAM population. There were broad variations in Zadoks’ scores and plant height measurements (Table 1). Despite the small size of the bi-parental families within the OzNAM population, Zadoks’ score and plant height measurements showed phenotypic variation within and between individual bi-parental families across different trials (Appendix A). Phenotypic variation could also be seen between the Gladius- and Scout-derived sub-populations. Even though we culled and selected plants based on maturity and plant height, substantial genetic diversity for these traits was preserved within the NAM. The OzNAM was designed to complement the Australian collection of multi-parental populations and the culling and selection succeeded in maintaining the plant height and maturity within the range that is suitable for it to be evaluated in the target environments.

SNP markers have been extensively utilized in QTL mapping and plant breeding [39]. While these markers have enabled great strides to be made in QTL mapping, SNP technologies that enable only bi-allele calls cannot reveal multi-allelic variation. Multi-allelic QTL exist in germplasm populations and plant breeding procedures aim to reassemble and accumulate many allelic variations in germplasm [40]. Furthermore, the multi-parent property of NAM populations allows for the detection of QTL with multiple alleles [41]. Therefore, to efficiently detect a QTL and have comprehensive genetic information about that QTL, use of multi-allelic markers is more relevant than using SNP markers that can only detect two alleles per locus. In the present study, we made use of multi-allelic markers and were able to reveal the diversity of the population and to detect QTL with their multiple alleles.

The haplotype genotyping data and population structure of the OzNAM showed high genetic diversity among populations and parents. As expected, the NAM families clustered into two distinct groups of Scout- and Gladius-derived families (Figure 4). Furthermore, within the Scout- and Gladius-derived family clusters, RILs from the same exotic parents also grouped together showing that the OzNAM population is a fusion of two NAM populations (Gladius and Scout NAM populations) which in themselves are composed of multiple bi-parental RIL families. While other wheat multi-parental populations have incorporated genetic diversity as much as possible, none has as many diverse parents as the OzNAM and, to the best of our knowledge, none have utilised multi-allelic markers that captures multi-allelic variation that exist within the population. The multi-allelic genotyping technology employed in the OzNAM reveals more diversity than that revealed by the SNP technology that has been employed in many other multi-parent populations.

NAM populations require carefully designed statistical models to appropriately represent the underlying genetic architecture of QTL. NAM populations have been analysed using either the principle of association mapping or linkage analysis [17,42,43,44,45,46] and most of these procedures do not accommodate multi-allelic marker data. For example, the method which was adopted in a number of maize NAM studies, the Joint Inclusive Composite Interval Mapping (JICIM) method, though designed to be an efficient and specialty method for the joint QTL linkage mapping of NAM populations [43], does not support the multi-allelic nature of the NAM population. Association mapping methods for analysing those populations have also the problem of a high false negative rate, which is due to the use of a stringent experiment-wise multiple testing correction criterion.

The RTM-GWAS method was developed to overcome these limitations [47]. This method supports multi-allelic data, uses a two-stage multi-locus multi-allele model and has a built-in experiment-wide criterion for correction of multiple testing [48]. In soybean NAM populations, RTM-GWAS could detect almost all the QTL that were detected by other methods with multiple alleles [41]. Here, we used the RTM-GWAS to detect QTL associated with maturity and plant height in a wheat NAM population. We identified five QTL with their multiple alleles for Zadoks’ score and three QTL with their multiple alleles for plant height. Maturity and plant height in wheat are traits that are mostly affected by major genes and this explains why we could detect few QTL for these traits.

Despite the relatively small size of our population, we could demonstrate the utility of our NAM population in QTL mapping. We could detect QTL within known genes that are already in use in marker assisted breeding [27]. In wheat, maturity is controlled by genes that fall into three categories, which are (i) photoperiod response genes (*Ppd*), (ii) vernalisation response genes (*Vrn*), and (iii) earliness per se genes (*Eps*) [49]. Photoperiod response is important for adaptation of wheat to various environments. Responsiveness to photoperiod varies in wheat varieties, with some varieties being insensitive to photoperiod. Photoperiod sensitive varieties flower early under long day conditions while the photoperiod-insensitive varieties flower at a similar time regardless of day length [32]. The QTL for Zadoks’ score located on chromosome 2D is analogous to the photoperiod-response gene *Ppd-D1* [32]. The *Ppd-D1* QTL was detected in all four trials (Table 2) and is known to be one of the strongest genes affecting photoperiod response in wheat [50]. A *Vrn* gene was previously mapped to a region on chromosome 5A [51] while an *Eps* gene was previously mapped to a region on chromosome 3A [52]. We can speculate that Zadoks’ score QTL on chromosome 5A and 3A could be associated with *Vrn* and *Eps* genes, respectively.

The semi-dwarf genes *Rht-B1* and *Rht-D1* were introduced in wheat varieties in the 1960s. Alleles from both loci produce similar increases in wheat productivity and reductions in plant height [53]. When a wheat genotype carries semi-dwarf alleles at both loci, it results in an extreme phenotype called double dwarf. Plant height QTL detected on chromosomes 4B and 4D correspond to the *Rht-B1* and *Rht-D1* [54] genes, respectively. *Rht-B1* was only detected in the 2017 trial while *Rht-D1* was detected in all the trials. Both the Gladius and Scout reference parents carry the dwarfing allele at *Rht-D1* and it is expected that most of the families in the NAM will be segregating for this gene thus its detection in all the trials. On the other hand, only a number of exotic lines carry the dwarfing allele at *Rht-B1* and so only a few families will be segregating for this gene. Failure to detect the *Rht-B1* QTL in subsequent trials indicates that selection for plant height may have eliminated or reduced the frequency of the dwarfing allele at this locus.

In the OzNAM, individual lines carry up to 61.5% of exotic alleles and up to 42% of individual chromosomes is represented by alleles from exotic parents (Appendix A). The distribution of various exotic alleles across the genome in OzNAM is indicative of the genetic variation that has been contributed by the exotic parents. Differences in the amount and distribution of exotic alleles between chromosomes and subpopulations of our NAM indicates that patterns of introgression are not conserved across the genome and among subpopulations. During the development of the NAM, backcrossing the F_1_ progenies to their respective reference parents and culling ensured that we retrieved more of the reference parent genetic background and eliminated some of the undesirable traits from the exotic parents. This was the case with maturity and plant height. Scout and some of the exotic parents carry the photoperiod insensitive allele while Gladius carries the photoperiod sensitive allele. Selection of OzNAM lines based on their maturity slightly favoured the photoperiod-insensitive alleles and we can see a higher frequency (0.68) of exotic alleles around the *Ppd-D1* locus in the Gladius subpopulation while we see the opposite in the Scout subpopulation. Both Scout and Gladius carry the dwarfing allele at the *Rht-D1* locus and the wild-type tall allele at *Rht-B1* locus. Culling to eliminate double dwarf genotypes in the NAM might be responsible for the lower frequency of exotic dwarfing alleles around the *Rht- B1* locus in both subpopulations.

In terms of identifying lines useful to breeding programs, the phenotypic data from the OzNAM subset revealed at least three lines in three trials that exhibit higher yield compared to the reference parents and some elite varieties (checks) that are already on the Australian market. Our field data indicated that the selection we imposed on the population based on maturity and plant height was successful and still maintained diversity in the population. However, we observed some lines with Zadoks’ scores outside the selected range in Dandaragan, which could be because all selections were made based on data collected from Roseworthy trials only. The differences in day length, temperature, rainfall and sowing dates among the sites could have been some of the reasons for the differences in maturity of the NAM lines from one site to the other.

The OzNAM population has aided the use of exotic germplasm and it is available as a community resource for wheat genetics and breeding programs in Australia and around the world. The 75 parental lines included in the OzNAM compared to twenty-six [55] and fifty-two [38] in other wheat NAM populations suggests the presence of potentially more diversity, more rare alleles, more recombination events and more recombinant haplotypes, which in turn translates to a greater representation of lines with favourable allelic combinations [18]. Moreover, the two-reference parent design makes comprehensive trait dissection easier by correlating the findings between families that share the same exotic parent. Many QTL mapping studies validate their results by checking the correspondence of identified QTL with previously identified QTL. With the OzNAM QTL, findings can be validated within the population, thus improving the adoption of these results in breeding.

## 4. Materials and Methods

### 4.1. Plant Material

To develop the OzNAM population, 76 diverse exotic donor lines were selected from a diversity panel described in [27]. Figure 5a shows the diversity present in the diversity panel from which the 76 diverse exotic parents were selected. The diverse donor lines were selected to represent diverse exotic germplasm with terminal drought and heat tolerance as well as nitrogen use efficiency in wheat. Forty-three donor lines were selected for diversity and originating from countries with dry and hot weather conditions, 22 lines were selected for known tolerance to heat and 11 for nitrogen use efficiency (Figure 5b, Appendix A). To enable the development and trait evaluation of the NAM population under Australian agronomic conditions, two Australian modern wheat varieties, Gladius (RAC-875/KRICHAUFF//EXCALIBUR/KUKRI/3/RAC-875/KRICHAUFF/4/RAC-875//EXCALIBUR/KUKRI, Australian Grain Technologie—AGT) and Scout (SUNSTATE/QH-71-6//YITPI, LongReach Plant Breeders) were selected as the common reference parents. Gladius and Scout are locally adapted varieties, with medium season maturity, good grain size and are high yielding.

### 4.2. Population Development

The OzNAM population was developed by crossing each of the exotic donor lines to Gladius and Scout. The resultant F_1_ plants from each exotic donor by Gladius and Scout were backcrossed to Gladius and Scout to create the BC_1_F_1_ generation, which was then advanced to BC_1_F_4:6_ by single seed descent (Figure 6). Culling for extreme maturity and double dwarves was done in the BC_1_F_2_ generation to eliminate plants that would be hard to phenotype in the field. The BC_1_F_4:6_ lines (or recombinant inbred lines, RIL) were used for phenotyping in multi-environmental trials. For easy nomenclature, the exotic parents were coded EP01 to EP76 and each NAM family was named after the reference parent (GL: Gladius and SC: Scout) and the corresponding exotic donor parent code name, e.g., the family developed between Gladius and exotic parent 01 was designated as the GLEP01 family.

### 4.3. Genotyping of the NAM Population

Leaf tissue from the NAM BC_1_F_4_ lines and the NAM parents was collected; freeze-dried and genomic DNA was isolated. In preparation for targeted genotyping-by-sequencing (tGBS), DNA samples were diluted to 100 ng mm^−3^.

#### 4.3.1. Library Preparation

A custom tGBS assay based on molecular inversion probes (MIPs) (CustomArray, Bothell, WA, USA) was designed to target 12,179 SNPs chosen from the iSelect wheat 90K Infinium SNP array [56] to provide genome-wide coverage. In brief, about 400 ng of purified gDNA was hybridised with 5 ng of MIP probes in 10× Ampligase buffer (Astral Scientific, Gymea, NSW, Australia) by denaturation at 95 °C for 10 min in a thermocycler, followed by a gradual decrease in temperature (0.1 °C/s) to 55 °C and incubation at 55 °C for 2 h. To circularize MIP probes correctly hybridised to targeted SNP loci, 3 µL of gap-fill mix was added to the reaction to provide a final concentration of 200 µM dNTPs, 1 U of Phusion Hot Start II DNA polymerase (ThermoFisher, Waltham, MA, USA) and 10 U of Ampligase DNA ligase (Astral, Australia) in 10× Ampligase buffer, before incubation at 55 °C for 1 h, followed by 72 °C for 15 min. Next, non-circularized MIP probes were removed by adding 1 U of NotI-HF (NEB, Ipswich, MA, USA) to the reaction, followed by incubation at 37 °C for 1 h and 94 °C for 30 s, and then 5 µL of exonuclease mix containing 4 U of exonuclease I (NEB, Ipswich, MA, USA), 18 U of exonuclease III (NEB, Ipswich, MA, USA), 4 U T7 exonuclease (NEB, Ipswich, MA, USA), 0.4 U exonuclease T (NEB, Ipswich, MA, USA), 3 U RecJf (NEB, Ipswich, MA, USA) and 0.2 U lambda exonuclease (NEB, Ipswich, MA, USA), followed by incubation at 37 °C for 2 h, 80 °C for 10 min and 95 °C for 5 min. The circularized MIP probes were PCR amplified to generate products compatible with Illumina next-generation sequencing platforms. PCR reactions were set up in 50 µL reaction with 11.5 µL circularized MIP probes, 0.5 µL Phusion HF DNA Polymerase (2U/µL) (ThermoFisher, Waltham, MA, USA), 10 µL 5× Phusion HF PCR buffer (ThermoFisher, Waltham, MA, USA), 2 µL combinatorial PCR index primers at 2 µM, 0.4 µL of 25 mM dNTP mix, 1 µL 0.001× SYBR green mix, and 24.6 µL H_2_O. PCR was performed with initial denaturation at 98 °C for 30 s, followed by 24 cycles of 3-step PCR starting at 98 °C for 30 s and then annealing/extension at 63 °C for 20 s and 72 °C for 30 s on a CFX96 Touch Real-Time PCR machine (Bio-Rad, Hercules, CA, USA). The amplified products were pooled in equimolar amounts for each sample based on qPCR quantification, prior to purification using the Agencourt AMPure XP SPRI beads (Beckman Coulter, Lane Cove, NSW, Australia) according to the manufacturer’s instructions. The pooled libraries were titrated using the KAPA quantification kit (MilliporeSigma, Burlington, MA, USA) and sequencing was performed on the Illumina HiSeq platform (Illumina, San Diego, CA, USA) using 384-plex custom sequencing primers according to the manufacturer’s instructions.

#### 4.3.2. tGBS Allele Calling and Data Analysis

A custom approach was developed to call genotypes from the tGBS data in order to maximise the potential for detecting novel polymorphism. Raw paired-end sequence reads were de-multiplexed and trimmed to remove Illumina adapter sequences using a custom script before being overlapped to form single reads using PEAR software v0.9.8 [57]. The overlapped reads generated for each of the 78 parental lines were used to generate an allele specific reference (ASR) using custom scripts. A minimum of four identical reads was required within each sample to form an ASR and all the ASRs generated from each parent were combined. The physical position of each ASR in the wheat genome was obtained by aligning its sequence to the reference genome sequence assembly for bread wheat cultivar Chinese Spring (IWGSC RefSeq v1.0) [58] using the Nuclear software v3.6.16 (GYDLE Inc. Montreal, Canada, http://www.gydle.com (accessed on 15 March 2021)). ASRs mapping to the same position in the genome were considered allelic. Genotypes were called using custom scripts after aligning the overlapped sequence reads for each sample to the physically mapped ASRs. Overall, a total of 16,439 SNPs and 418 INDELs were called.

#### 4.3.3. Linkage Disequilibrium Analysis and Data Imputation

To generate the multi-allelic haplotype markers, raw genotype data were filtered to retain only polymorphic markers among the parents. The genotype files (VCF version 4.2) were split according to Scout or Gladius genetic background before the linkage disequilibrium (LD) analysis was performed using PLINK v1.9 [59]. A list of SNPs in LD with the targeted SNPs was generated using the show-tags function in PLINK with the parameters r^2^ = 0.7, minor allele frequency = 0.01, missing genotype = 0.06 and within 10,000 kilobases of the target. A total of 12,464 and 13,104 SNPs were tagged in the Scout and Gladius genotypic data, respectively. Codominant SNPs with >40% missing values and dominant SNPs with >20% missing values were excluded. Missing data for the remaining SNPs were imputed using LinkImpute version 1.1.5 with default parameters [60].

### 4.4. Phenotyping and Statistical Analysis of the NAM Population

Due to the size of the population and the constraints in resources to evaluate the entire NAM population in a single trial, a subset of the BC_1_F_4:6_ NAM RIL families consisting of a total of 530 lines from 28 families with 8–26 lines per family being selected and grown under rainfed conditions in a field trial located near Roseworthy (34°31′18.9′′ S 138°39′43.9′′ E) in South Australia during the 2017 growing season. In total, 400 NAM RILs and 15 check varieties were grown with two replicates and 130 NAM RILs were grown with one replicate in a partially replicated design [61]. The field trial consisted of 960 plots arranged in 40 rows by 24 ranges. Each plot consisted of six rows with 22.5 cm spacing between rows and a length of 4 m. Phenotypic data on maturity, plant height and yield were collected. Although BC_1_F_2_ plants that showed extreme maturity or double-dwarf stature were culled, some lines still showed these phenotypes, which strongly affect yield and biased the dataset. Based on the phenotypic data collected in this first year of field evaluation, we chose the NAM subset that had (i) maturity between two local varieties, Axe (early maturing) and Yitpi (late maturing), (ii) plant height below 115 cm and (iii) higher grain yield in addition to the first two conditions. A total of 238 lines from 15 families with 11–22 lines per family and 14 check varieties were selected and grown at two sites in 2018 (Roseworthy: 34°30′54.3′′ S 138°40′06.6′′ E and Dandaragan: 30°41′07.1′′ S 115°39′54.4′′ E) and one site in 2019 (Roseworthy: 34°30′51.4′′ S 138°41′23.9′′ E). The Roseworthy trials were sown on the 18th of May of each year in a randomized complete block design (RCBD), consisting of 504 plots arranged in 21 rows by 24 ranges. The Dandaragan trial was sown on the 7th of June 2018 using a RCBD, consisting of 504 plots arranged in 42 rows by 12 ranges. Temperature, rainfall and other environmental data for each trial was collected from a station not more than 5 km away from each trial site (Appendix A).

Maturity was estimated by noting the growth stage using the Zadoks’ scale [31] at about 110 days after sowing. Plant height was measured in centimetres from the soil surface to the tip of the spike excluding awns for three randomly chosen plants per plot at physiological maturity. The final height measurement was the average of the three measurements per plot. Maturity in the 2017 field trial was estimated from 530 NAM RILs and 16 check varieties, while plant height was estimated from 495 NAM RILs and 16 check varieties due to lodging in some of the RILs at physiological maturity when height measurements were taken. In the 2018 and 2019 trials, Zadoks’ scores and plant height were estimated from 238 NAM RILs and 14 check varieties. Plant height data were not collected for the Roseworthy 2018 field trial.

The phenotypic data from the four field trials were analysed separately by modelling spatial trends to obtain the best linear unbiased predictors (BLUPs) of random effects using the R package SpATS [62]. Genotypes and experimental design factors were fitted as random effects. Heritability was also calculated in SpATS as the generalized heritability proposed in [63]. BLUPs for each trait were used as the phenotypic values in the GWAS.

### 4.5. Introgression and Population Structure

The frequency of exotic alleles in the NAM was estimated separately for Gladius- and Scout-derived sub-populations. Homozygous polymorphic markers in each subpopulation were identified and used to calculate the percentage of lines within each subpopulation that carried the exotic alleles at each marker locus. The calculated frequency of exotic alleles was reported for each chromosome and line. Heat maps were developed in R using the chromoMap package [64] to show the proportion of exotic alleles within each subpopulation for each chromosome. Markers proxy to the *Ppd-D1*, *Rht-B1* and *Rht-D1* loci were used to infer the pattern and level of Introgression around these loci. The population structure for the whole NAM population, including the parents, was inferred from the multi-allelic haplotype marker-based genetic similarity coefficient (GSC) matrix constructed using a restricted two-stage multi-locus multi-allele genome-wide association study (RTM-GWAS) implemented in the RTM-GWAS software [48]. The GSC between two individuals is defined as the proportion of loci that are in identity-by-state [48,65]. The multi-allelic haplotype genomic markers for 3610 individuals were filtered to remove markers genotyped in <70% of the samples and those with heterozygosity >6%. In total, 498 RILs that had less than 50% genotyped markers were eliminated. After this filtering process, the remaining set of 4964 markers and 3112 individuals was used for population structure analysis. The inferred population structure was visualised in a scatterplot of the top eigenvectors of the GSC matrix using the R package ggplot2 [66].

### 4.6. QTL Mapping in the NAM Population

To assess the potential of the NAM population in QTL mapping, we used a subset (530 RILs) of the NAM population and 4615 multi-allelic haplotype markers. The filtering of the markers was the same as mentioned above. The RTM-GWAS procedure [48], which supports multi-allelic haplotype marker data, was used for GWAS. The GWAS was carried out in a two-stage multi-locus procedure, as described in [48] and used the top 10 eigenvectors of the GSC matrix as covariates to correct for population structure. A significance level of *p* ≤ 0.0005 was applied at both stages of the analysis. Despite the incorporation of the built-in experiment-wide stepwise regression criterion that corrects for multiple testing, the Bonferroni correction was also used. Seven KASP markers—KASParMAS038, KASParMAS039, KASParMAS051, KASParMAS055, KASParMAS057, GPC_functional and AX-94529403_VrnA3—were added to the marker data for the GWAS. KASP markers KASParMAS038 and KASParMAS039 assay the plant height genes, *Rht-B1* [33] and *Rht-D1* [33], respectively, while KASP marker KASParMAS051 assays the photoperiod response (*Ppd-D1*) gene [32]. KASParMAS055 and KASParMAS057 assay the vernalisation gene *Vrn-A1* (Yan et al., 2004) while AX-94529403_VrnA3 assays the vernalisation gene *Vrn-A3.* The GPC_functional is a marker targeting the grain protein content locus, *GPC-B1* [67]. Together, the marker data and phenotypic BLUPs were used to detect QTL associated with maturity and plant in each trial.

## Figures and Tables

**Figure 1 ijms-22-04348-f001:**
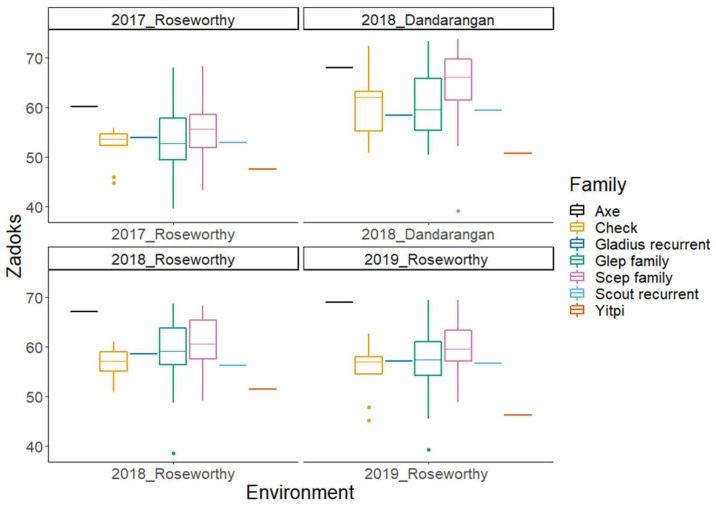
Boxplots of Zadoks’ score (maturity) BLUPs in four different environments for the subset of the Wheat Hub NAM population and the check varieties. The NAM subset population is separated into Gladius and Scout families. BLUPs for Axe and Yitpi represent the selection thresholds for Zadoks’ score.

**Figure 2 ijms-22-04348-f002:**
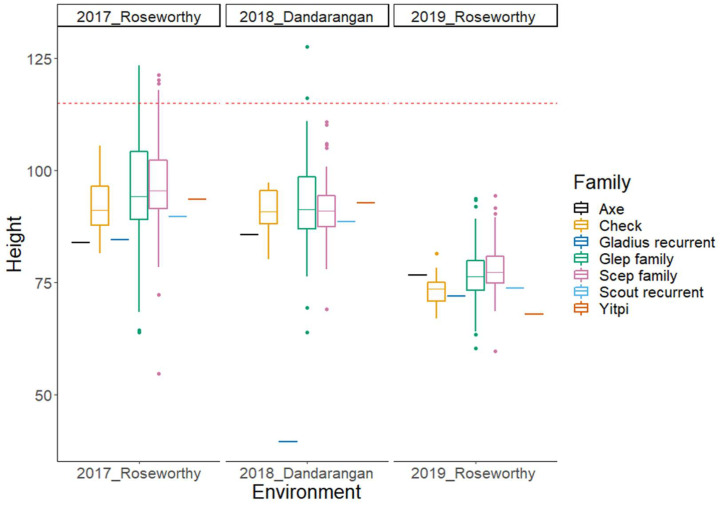
Boxplots of plant height BLUPs for the subset of the Wheat Hub NAM population and the check varieties in three different environments. The NAM subset population is separated into Gladius and Scout families. The horizontal dotted line represents the selection threshold of 115 cm for plant height.

**Figure 3 ijms-22-04348-f003:**
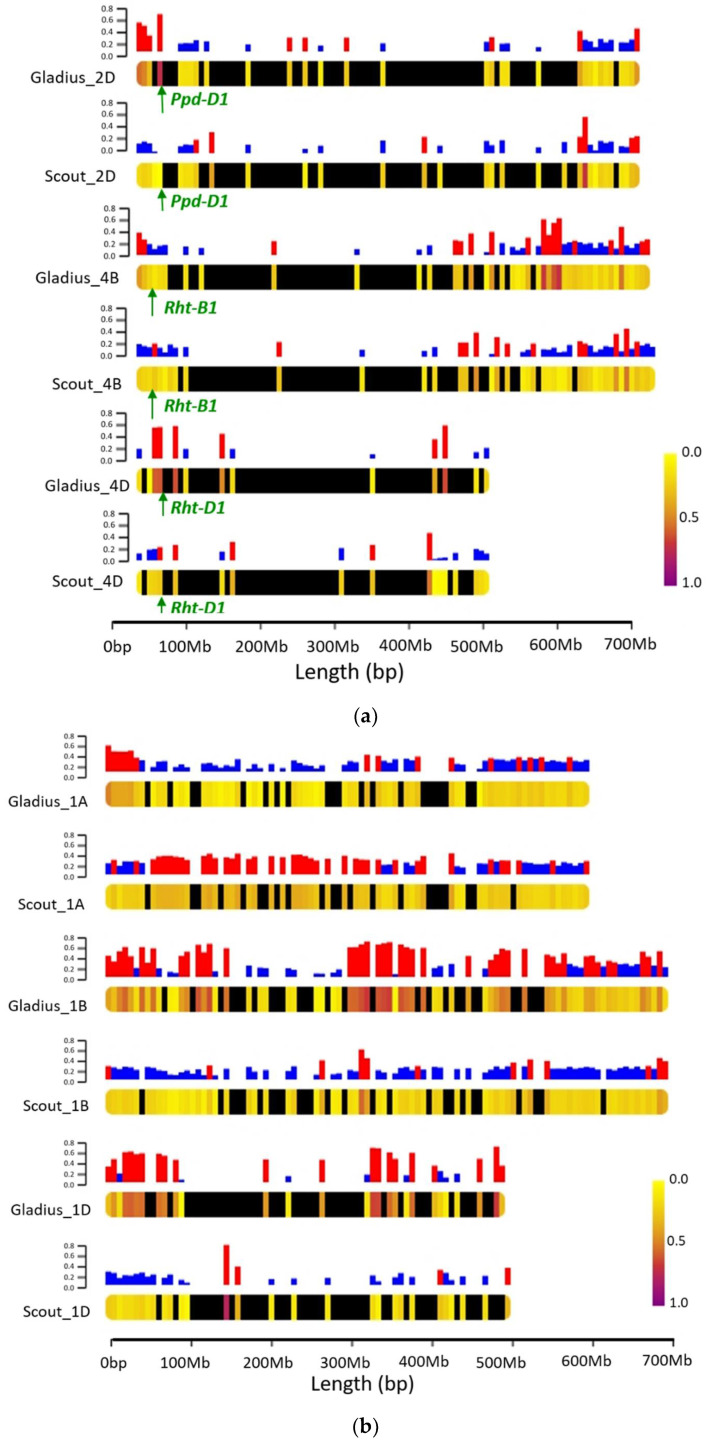
Chromosome heat maps showing proportions of exotic and reference parent alleles on different chromosomes: (**a**) Chromosomes 2D, 4B and 4D. The loci for relevant genes are indicated; (**b**) Chromosome 1. Bars on top of each chromosome represent the frequency of exotic alleles at each locus. Bars in red represent exotic allele frequencies greater than or equal to 0.25. Black segments on the chromosomes show regions of the chromosome with no markers in the data set.

**Figure 4 ijms-22-04348-f004:**
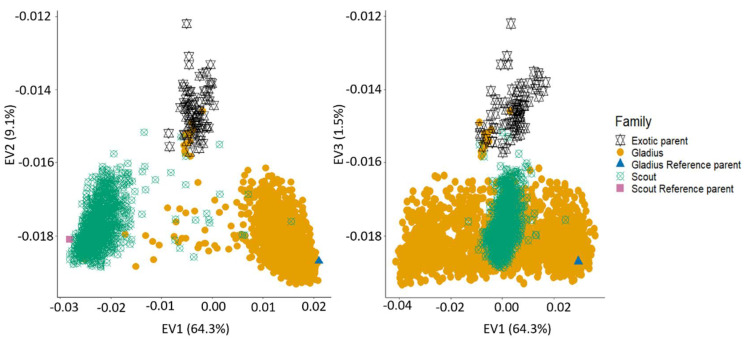
Population structure of the OzNAM population and their parents. Percentages on each axis refer to the proportion of variation explained by each eigenvector.

**Figure 5 ijms-22-04348-f005:**
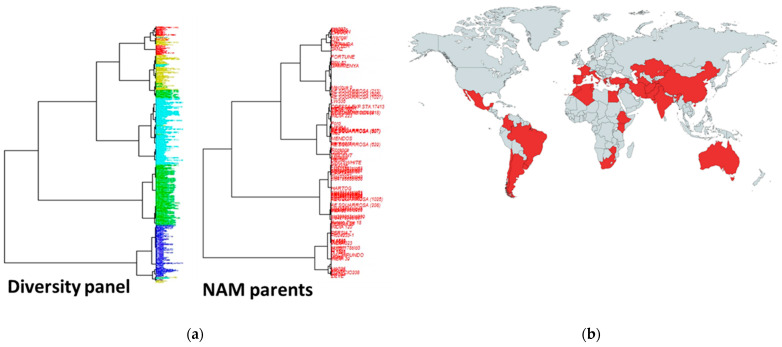
Characteristics of the NAM exotic parents (**a**) Phylogenic trees showing diversity in the diversity panel from which the diverse exotic parents were selected. The NAM parents are highlighted from the phylogenic tree; (**b**) geographical distribution of the NAM exotic parents.

**Figure 6 ijms-22-04348-f006:**
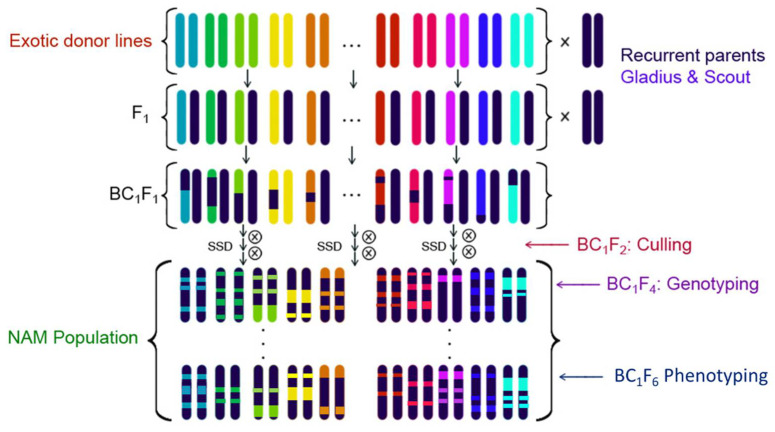
Crossing design for the development of OzNAM.

**Table 1 ijms-22-04348-t001:** Descriptive statistics and heritability estimates of the NAM subset evaluated in each of the four field trial environments.

Trait	Trial	Min	Max	Mean	N	SD	H^2^
Zadoks’ score	Roseworthy 2017	40	68	54.0	546	5.7	0.93
	Roseworthy 2018	39	69	60	252	5.2	0.93
	Dandaragan 2018	39	74	62	252	6.7	0.8
	Roseworthy 2019	39	70	59	252	5.9	0.91
Height (cm)	Roseworthy 2017	55	123	97	511	10.4	0.95
	Dandaragan 2018	40	128	92	252	8.6	0.81
	Roseworthy 2019	60	94	77	252	5.8	0.9

Min = Minimum, Max = Maximum, N = population size, SD = standard deviation, H^2^ = heritability

**Table 2 ijms-22-04348-t002:** QTL associated with maturity and plant height detected in four trial environments.

Trait	Environment	Marker	Chromosome	Number of Alleles	Corresponding Gene	*p*-Value	R^2^ (%)
Zadoks’ Score (maturity)	2017_Roseworthy	LDB_2D_1	2D	2	*Ppd-D1*	1.56283 × 10^−46^	34.3
		LDB_3A_624535417	3A	5	-	1.01741 × 10^−5^	3.5
		LDB_2B_58324935	2B	3	-	3.80867 × 10^−5^	3.2
	2018_Dandarangan	LDB_2D_1	2D	2	*Ppd-D1*	4.09762 × 10^−16^	26.9
		LDB_3A_510690367	3A	3	-	1.57639 × 10^−5^	7.1
	2018 Roseworthy	LDB_2D_1	2D	2	*Ppd-D1*	1.26164 × 10^−15^	28.5
		LDB_5A_1	5A	3	-	1.29529 × 10^−5^	9.4
	2019_Roseworthy	LDB_2D_1	2D	2	*Ppd-D1*	7.46238 × 10^−20^	34.1
Plant Height	Roseworthy 2017	LDB_4D_1	4D	2	*Rht-D1*	2.23596 × 10^−49^	39.6
		LDB_4B_1	4B	2	*Rht-B1*	2.80 × 10^−25^	13.6
	Dandaragan 2018	LDB_4D_1	4D	2	*Rht-D1*	1.3935 × 10^−14^	27.1
		LDB_6B_679622060	6B	11	-	7.20103 × 10^−13^	18.6
	Roseworthy 2019	LDB_4D_1	4D	2	*Rht-D1*	1.58475 × 10^−11^	21.9

R^2^ = percentage variation explained.

## Data Availability

Raw phenotypic and genotypic data used in this study are available from Figshare (DOI: 10.25909/12971918)

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
