# Peer review of "Development of an Australian Bread Wheat Nested Association Mapping Population, a New Genetic Diversity Resource for Breeding under Dry and Hot Climates"

_ijms, 2021, doi:10.3390/ijms22094348_

Round 1
Reviewer 1 Report
The study of Chidzanga et al. is well-written, and includes excessive effort and experimental data. The paper centers around the genetic background of two traits (maturity and plant height) in wheat. However, the importance of these two traits and their relation with yield is not apparent.
(1) In the Abstract and Introduction sections, a lot of attention (and text) is devoted to yield loss owing to drought and heat stress. However, the authors neither measured yield, nor imposed either of the two abiotic stressors. Instead, they select parents encompassing tolerance to them (371–372). Therefore, is the current study really a step towards enhanced drought and heat stress tolerance (as highlighted in the title)?
(2) Four field trials were performed at three sites [Roseworthy (two sites) & Dandaragan] and three years (2017, 2018, 2019). You need to provide a table depicting tested lines, site, sowing and evaluation time (month, year, date), cultivation measures, environmental conditions during plant growth (air temperature, relative air humidity, daily light integral, photoperiod, rainfall; lines 483–484), traits under study. This is the only readable and understandable way for the audience.
(3) Authors assessed maturity and plant height. Why did you assess these two traits? It needs to be explicitly mentioned in both the introduction and discussion sections. Scattered information is provided in the discussion, and this brings repetitions.
(4) Maturity is affected by photoperiod (line 288). Then we realize that maturity is also influenced by two more factors (lines 325–328). These sections ought to be merged.
(5) Is flowering time synonymous to maturity? Day length, temperature and sowing dates account for differences in the flowering time among trials (lines 353–354). What other effect of sowing date is expected, other than light period and temperature? Do you mean water availability? Is there something else?
(6) Lines 54–55: consider the studies of Carvalho et al. 2015 (Mol. Breeding 35, 172) and Fanourakis et al. 2015 (Ann. Bot. 115, 555–565) dealing with QTLs associated with stomatal closing ability.
(7) factors affecting plant height (lines 300–303, 334–338). This again brings repetitions. You need to provide these together.
(8) Line 476: screening and selection was based on grain yield. But nowhere is mentioned that grain yield was assessed. These data are very valuable.
Author Response
Revisions requested ijms-1183005
Dear editor,
We have revised the manuscript as per recommendations made by reviewers and we would like to thank you and the reviewers for the valuable feedback provided. Please find below our response to reviewers’ comments.
We hope you find this new version suitable for publication.
With best regards,
Melissa Garcia
Reviewer 1
(1) In the Abstract and Introduction sections, a lot of attention (and text) is devoted to yield loss owing to drought and heat stress. However, the authors neither measured yield, nor imposed either of the two abiotic stressors. Instead, they select parents encompassing tolerance to them (371–372). Therefore, is the current study really a step towards enhanced drought and heat stress tolerance (as highlighted in the title)?
Response: We agree with the reviewer and we have edited the text to shift the attention from yield loss owing to drought and heat stress.
Abstract:
Lines 23-26 deleted beginning of the sentence and revised it.
Lines 38-40 rephrased the sentence.
Introduction:
Lines 87-94 Deleted the initial sentence and added new text. Although we did not impose heat and drought, these occur naturally in Australia.
(2) Four field trials were performed at three sites [Roseworthy (two sites) & Dandaragan] and three years (2017, 2018, 2019). You need to provide a table depicting tested lines, site, sowing and evaluation time (month, year, date), cultivation measures, environmental conditions during plant growth (air temperature, relative air humidity, daily light integral, photoperiod, rainfall; lines 483–484), traits under study. This is the only readable and understandable way for the audience.
Response: We thank the reviewer for this suggestion. The table was added to supplementary data as Table S3: Weather data and setup of the field trials. The table is mentioned in the manuscript on line 534.
(3) Authors assessed maturity and plant height. Why did you assess these two traits? It needs to be explicitly mentioned in both the introduction and discussion sections. Scattered information is provided in the discussion, and this brings repetitions.
Response: Because we would like to demonstrate the usefulness of the population in QTL mapping, it was important to look at traits for which there were known strong effect loci as well as other smaller effect loci. Both maturity and plant height fit this criterion and are selected in breeding due to their correlation with yield. We have adjusted the text as suggested, and we hope the changes make it clearer for the reader.
Introduction: Lines 116-123 Added new text to justify the use of maturity and plant height.
(4) Maturity is affected by photoperiod (line 288). Then we realize that maturity is also influenced by two more factors (lines 325–328). These sections ought to be merged
Response: We agree with reviewer’s suggestion and we merged the text from line 305 -309 with text in lines 345-350.
(5) Is flowering time synonymous to maturity? Day length, temperature and sowing dates account for differences in the flowering time among trials (lines 353–354). What other effect of sowing date is expected, other than light period and temperature? Do you mean water availability? Is there something else?
Response: Flowering time was used as synonymous to maturity in the manuscript. We agree with the reviewer and we have replaced flowering time with maturity throughout the text.
We also included rainfall as another factor that could affect maturity (line 397).
(6) Lines 54–55: consider the studies of Carvalho et al. 2015 (Mol. Breeding 35, 172) and Fanourakis et al. 2015 (Ann. Bot. 115, 555–565) dealing with QTLs associated with stomatal closing ability.
Response: We thank the reviewer for the suggestion. We have considered these studies that were done in Roses and Tomatoes but we have decided to keep our text and reference as it is.
(7) factors affecting plant height (lines 300–303, 334–338). This again brings repetitions. You need to provide these together.
Response: We agree with the reviewer and we have moved text from lines 314-317 and merged it with text in lines 357-360. We also deleted sentence in line 357 to avoid repetition after the merge.
To improve the flow after these changes, we have also moved text from lines 291-316 to lines 366-383.
(8) Line 476: screening and selection was based on grain yield. But nowhere is mentioned that grain yield was assessed. These data are very valuable.
Response: We agree with the reviewer that these data are very valuable, and we are working on the data at the moment. Analysing yield data from the whole population evaluated in different years and environments and presenting it clearly are complex tasks and we believe we need a separate manuscript to do so. In the present manuscript, we want to highlight this population as a resource for research and breeding and present it to the community.
Reviewer 2 Report
Dear Autors,
Reviewer comments ijms-1183005
The manuscript entitled „Development of an Australian bread wheat nested association mapping (NAM) population, a new genetic diversity resource for breeding under dry and hot climates“ represents a useful study on the development of nested association mapping (NAM) population derived from two modern Australian cultivars Gladius and Scout and its utilization for mapping of QTLs associated with plant maturity determined as Zadoks scale and plant height from the plants grown at two Australian locations (Roseworthy, Dandarangan) during three years (2017-2019).
The manuscript provides detailed description of the development of nested association mapping (NAM) population for QTL mapping as an alternative approach to classical biparental populations (populations derived from a cross between two parental genotypes with contrasting featurs, e.g., tolerance to environmental stress) as well as to genome-wide association mapping (GWAS) approach based on QTL mapping in the large collections of unrelated genotypes comprising modern cultivars and breeding materials as well as landraces and exotic germplasm, i.e., genotypes with contrasting features.
The manuscript provides detailed information on the NAM population development as well as the results on the mapping of QTLs associated with plant height and maturity in the NAM population derived from two modern Australian wheat cultivars Gladius and Scout tested on two locations Roseworthy and Dandarangan during a three-year period (2017-2019).
I have no major comment on the present manuscript.
I have only a few formal comments which are given below:
Abstract, line 31: Correct the typing error in the term „varieties“ (not „variates“) in „two Australian elite varieties Gladius and Scout.“
Introduction, line 47: Add a space betwen the word „traits“ and the following reference.
Introduction, line 84: Add a comma prior to the word „respectively“ in the sentence „..in yield losses up to 44% and 53%, respectively…“
Results, line 125: Add a comma prior to the word „respectively“ in the sentence „… and plant height in different sites, respectively.“
Results, lines 157, 159: Add a comma prior to the word „respectively“.
Results, line 176: Add a comma following the words „For Gladius derived families,…“
Discussion, line 261: Add a space between the word „alleles“ and the following reference.
Discussion, line 263: Add a comma following the words „In the present study,…“
Discussion, line 274: Add a comma both preceding and following the words „to the best of our knowledge,…“
Discussion, line 289: Replace the word „in“ by „to“ in the sentence „…which is important for adaptation of wheat to various environments.“
Discussion, line 324: Remove the word „of“ following the word „Despite“ in the sentence „Despite the relatively small size of our population,…“
Discussion, line 334: Add a comma prior to the word „respectively“ in the sentence „…that Zadoks´ score QTL on chromosome 5A and 3A could be associated with Vrn and Eps genes, respectively.“
Materials and methods, line 402: Use SI units for volume, i.e., write „100 ng mm-3“ instead of „100 ng/μl.“
Materials and methods, line 416: Use „s“ insetad of „sec“ for the time unit „second“ in „30 s“.
Materials and methods, line 459: Add a comma prior to the word „respectively“.
Final recommendation: Accept after a minor revision.
Author Response
Revisions requested ijms-1183005
Dear editor,
We have revised the manuscript as per recommendations made by reviewers and we would like to thank you and the reviewers for the valuable feedback provided. Please find below our response to reviewers’ comments.
We hope you find this new version suitable for publication.
With best regards,
Melissa Garcia
Reviewer 2
(1) Abstract, line 31: Correct the typing error in the term „varieties“ (not „variates“) in „two Australian elite varieties Gladius and Scout.“
Response: done.
(2) Introduction, line 47: Add a space between the word „traits“ and the following reference.
Response: done, line 48
(3) Introduction, line 84: Add a comma prior to the word „respectively“ in the sentence „..in yield losses up to 44% and 53%, respectively…“
Response: This sentence was deleted to address one of the comments from reviewer 1.
(4) Results, line 125: Add a comma prior to the word „respectively“ in the sentence „… and plant height in different sites, respectively.“
Response: done, line 138
(5) Results, lines 157, 159: Add a comma prior to the word „respectively“.
Response: done, line 170 and 172
(6) Results, line 176: Add a comma following the words „For Gladius derived families,…“
Response: done, line 190
(7) Discussion, line 261: Add a space between the word „alleles“ and the following reference.
Response: done, line 277
(8) Discussion, line 263: Add a comma following the words „In the present study,…“
Response: done, line 279
(9) Discussion, line 274: Add a comma both preceding and following the words „to the best of our knowledge,…“
Response: done, line 290
(10) Discussion, line 289: Replace the word „in“ by „to“ in the sentence „…which is important for adaptation of wheat to various environments.“
Response: done, line 347
(11) Discussion, line 324: Remove the word „of“ following the word „Despite“ in the sentence „Despite the relatively small size of our population,…“
Response: done, line 341
(12) Discussion, line 334: Add a comma prior to the word „respectively“ in the sentence „…that Zadoks´ score QTL on chromosome 5A and 3A could be associated with Vrn and Eps genes, respectively.“
Response: done, line 356
(13) Materials and methods, line 402: Use SI units for volume, i.e., write „100 ng mm-3“ instead of „100 ng/μl.“
Response: done, line 447
(14) Materials and methods, line 416: Use „s“ instead of „sec“ for the time unit „second“ in „30 s“.
Response: done,line 461
(15) Materials and methods, line 459: Add a comma prior to the word „respectively“.
Response: done, line 506
Round 2
Reviewer 1 Report
my comments were partly addressed, based on the authors' convenince
no point to raise more comments